# Risk assessment and mitigation evaluation of future yellow fever outbreaks under different climate scenarios: Insight from a case study of Brazil

Tedi Ramaj[1]*, Xiaotian Wu[2], Marco Tosato[1], Fernando Morelli[3], Yael Thollot[3], Edith Langevin[3], Edward Thommes[4,5], Woldegebriel Assefa Woldegerima[1], Jianhong Wu[1]

1 Laboratory for Industrial and Applied Mathematics, Department of Mathematics and Statistics, York University, Toronto, Canada, 2 School of Science, Shanghai Maritime University, Shanghai, China, 3 Sanofi, Lyon, France, 4 Sanofi, Toronto, Canada, 5 Department of Mathematics and Statistics, University of Guelph, Guelph, Canada

* tramaj@yorku.ca

**Data availability statement:** All relevant data has been cited within the paper and may be

## Abstract

**Background:** Yellow Fever (YF), a disease typically transmitted to humans by infected mosquitoes, is endemic to regions such as South America. Climate change plays a crucial role in exacerbating the spread of YF.

**Model:** We formulate a mathematical model of YF transmission with a case study of the region in the southeastern Brazil with a well documented 2017/2018 outbreak. We validate the model using historical data, then run simulations to generate projections of future outbreaks under different climate scenarios in 2050. We also evaluate the outcomes of different mitigation measures such as emergency vaccination programs.

**Findings:** Our results suggest that under all projected climate scenarios, increasing temperatures will yield a marked increase in the total number of cases. Under RCP 8.5, the basic human infection reproduction number will increase by 11.4%, and the cumulative infections will increase by 8.1%. The model predicts a similar increase under a moderate radiative forcing scenario. The introduction of additional emergency vaccination, at a rate of 8.0% (equivalent to 60% vaccination coverage over the course of 15 weeks) of the susceptible population per week, can reduce this increase of cumulative cases to approximately 4.9%. This effect of emergency vaccines will be equivalent to alternative public health interventions to reduce the mosquito-to-human disease transmission effective contact by approximately 23.0%. Increasing temperatures and rainfall due to climate change are projected to increase YF cases. Vaccination can be an important part of integrative mitigating measures.

found in the corresponding references. Code has also been included in S1 Code for replication of results. Link to the data: https://sbim.org.br/images/files/informe-fa-21-11abr18-c.pdf.

**Funding:** This work was supported by the NSERC-Sanofi Alliance program in Vaccine Mathematics, Modelling, and Manufacturing (517504 to JW) and by the National Natural Science Foundation of China (12271346 to XW). The funders had no role in study design, data collection, and analysis, decision to publish, or preparation of the manuscript.

**Competing interests:** I have read the journal's policy and the authors of this manuscript have the following competing interests: FM, YT, EL, ET are employees of Sanofi and may hold stocks/shares.

## Author summary

Yellow Fever is a disease that can be transmitted between humans and mosquitoes, and its spread is often exacerbated by increasing temperature and precipitation patterns. We develop a mathematical model of Yellow Fever transmission in Southeastern Brazil to assess the long term impact of climate change on Yellow Fever transmission. Under the RCP 8.5 climate change scenario, in which temperature and rainfall are expected to increase, our model suggests that the basic reproduction number will increase by 11.4 % and the cumulative infections will increase by 8.1 % in this region. Our work also shows that the implementation of emergency vaccination, at a rate of 8.0 % of the susceptible population per week, in response to an outbreak, can help reduce the increase in cases by approximately 4.9 %.

## Introduction

Yellow Fever (YF) is a disease caused by a virus of the *Flavivirus* genus that is typically transmitted to humans (and non-human primates) via the bite of an infected mosquito, typically of the *Aedes* or *Haemagogus* genus [1–3]. YF symptoms can include fever, headache, myalgia, nausea, vomiting, diarrhea, with jaundice when there is hepatic involvement [4]. YF virus is endemic to tropical regions such as South America and Africa [5]. However, elevated temperatures related to natural and anthropogenic climate change may create more ideal conditions for mosquito development in previously temperate regions and contribute to the extension of YF—among other vector-borne diseases—into these regions [6]. It should also be noted that YF, along with other arboviruses, such as dengue, Zika, and chikungunya have increased significantly in the recent decades and expanded into new regions, including transmissions in North America and Europe [7,8]. Historically, YF epidemics have frequently occurred, resulting in severe illness and death [9]. Johansson et al. used a Bayesian model to estimate the probability of different infection outcomes by using data from 11 studies in Africa and South America between 1969 and 2011 [10]. They found that the probability of an individual infected with YF to experience severe disease was 12 % and the probability of death in individuals experiencing severe disease was 47 %. While the virus cannot be completely eradicated due to circulation in animal reservoirs, vaccination may be deployed to reduce the peak number of infections in a way so as not to burden medical resources [11]. Due to the inherent complexity of the prevention and control of YF and the global public health interest, interdisciplinary and integrative approach has been called for, including the World Health Organization's (WHO) project to Eliminate Yellow Fever Epidemics (EYE) [12]. Mathematical modelling can play a significant role in synthesizing data, knowledge and projections of ecological, epidemiological, environmental and behavioural conditions in the future to inform the preparedness and rapid response including the production, deployment and administration of vaccines.

Climate change has been attributed to various negative ecological impacts, and this is expected to continue if trends do not change. Representative concentration pathways (RCPs) are used to project and predict possible future climates. Taking temperature as an example, RCP 4.5 represents a moderate climate change scenario, with global temperatures increasing about 2°C globally, while the much harsher scenario RCP 8.5 represents global temperatures increasing about 4°C globally. As temperatures increase, previously temperate regions may become more favourable for mosquito development [13]. Regions currently endemic for YF

may also become more or less hospitable as increasing temperatures impact mosquito reproduction [14]. It is therefore important to integrate these projections into parameters, relevant to mosquito production, development, and mortality, of a comprehensive mathematical model for the transmission dynamics to assess the region-specific impact of climate changes on the YF public health burdens, an objective of our current study.

Wint et al. considered various hypotheses regarding why mosquitoes such as *Ae. aegypti* have not re-emerged in significant numbers in Europe [15]. While there are competing ideas on this topic, the authors mentioned that the most likely reason for the failure of mosquitoes to re-emerge is that the vectors typically require significant numbers to establish. The impact of climate change on regions like Europe was also considered by Liu-Helmersson et al., who found that under RCP 8.5 conditions, large parts of Southern Europe are in further danger of being invaded by *Ae. aegypti* [16]. Work by Kamal et al. has found similar patterns of distribution, under different RCP scenarios, in regions such as southern Canada and eastern Australia by 2050 and 2070 [17]. Monaghan et al. found that the RCP 4.5 pathway will yield 4 % less human cases of YF compared to the RCP 8.5 pathway, by 2060–2080 [18]. With the impact of climate change on YF dynamics in mind, it is important for any work on the topic to consider changing climate factors (i.e., temperature and rainfall) under which mosquitoes may be able to establish in previously temperate regions even under smaller initial numbers.

Our model formulation and analyses incorporate the impact of both temperature and rainfall on YF transmission dynamics into a standard epidemiological model for vector-borne disease transmission (Section 2). We make use of a compartmental SEAIR (Susceptible - Exposed - Asymptomatic Infectious - Symptomatic Infectious - Recovered) differential equation model to capture these dynamics. We calibrate the model using the 2017/2018 southeastern Brazil YF outbreak, which includes the regions of Rio de Janeiro and São Paulo, among others [19]. The choice of this site allows us to assess the impact of climate change on YF transmission in regions which are currently YF endemic zones. After the calibration and validation process, we analyse the basic human infection reproduction number (a qualitative metric for the outbreak potential and intensity) as a function of temperature and mitigation measures including the emergency vaccine coverage and the reduction of human-mosquito contacts for effective transmission. We then simulate administrating an emergency vaccine program to mitigate a future outbreak exacerbated by climate change. The incorporation of vaccination coverage in our model provides a novel and important contribution to the study of the impact of climate change on YF epidemics. This is because pharmaceutical measures can play a key mitigating role and quantifying their impact can be of significant importance to policy decision makers and public health officials. Our work evaluates the impact of introducing emergency vaccines (in endemic regions) in an integrative approach towards mitigating YF outbreaks under projected future climates.

## Methods

To model YF transmission dynamics, we make use of a standard vector-borne disease transmission dynamics model involving humans as hosts and mosquitoes as vectors. In the model, described in detail in S1 Appendix, humans are divided into seven non-overlapping compartments: susceptible ($S$), vaccinated susceptible ($V$), exposed ($E$), asymptomatic infectious ($A$), symptomatic infectious ($I$), recovered ($R$), and dead/removed ($D$); while mosquitoes are separated into three non-overlapping compartments: susceptible ($X$), exposed ($Y$), and infectious ($Z$). All model parameter values are either estimated, derived from the case study outbreak or taken from previously established literature.

Individuals in the $S$ compartment are susceptible and have no immunity against YF infection while the rest of the population initially has immunity due to "a combination of natural immunity or vaccination" [20,21]. Here, the vaccinated compartment, $V$, contains individuals who have received the *additional* emergency vaccine when the additional emergency vaccine is administered. The vaccination rate is given by $v$, and the vaccine efficacy is denoted by $\eta$. In subsequent sections, when we refer to emergency vaccination, we mean additional emergency vaccination administered on top of that given during the 2017/2018 outbreak.

Humans move to the exposed compartment, $E$, with mosquito-to-human transmission rate $\beta_1$. The fraction of infected humans that develop symptoms is given by $\phi$ and $1/\lambda_1$ is the human intrinsic incubation period. The recovery rate is given by $\gamma$, i.e., $1/\gamma$ gives the duration of infectiousness in humans. The mosquito development rate is given by $\delta(T, \zeta) = \delta_1(T)\delta_2(\zeta)$ and is dependent on the average temperature and average rainfall, $T$ and $\zeta$, respectively, assuming that the impact of temperature and rainfall on the development rate of mosquitoes are independent [22]. This approach captures the impact of both increasing temperatures and changing rainfall patterns on the development rates of mosquitoes. We use the Ricker function for the density-dependent reproduction rate, with the parameter $\sigma$ determined from the mosquito carrying capacity in the region under consideration. The human-to-mosquito transmission rate per infected human is given by $\beta_2$. The temperature-dependent parameters $\mu(T)$ and $1/\lambda_2(T)$ respectively represent the natural mosquito death rate and the extrinsic mosquito incubation period. The temperature and rainfall-dependent mosquito parameters are extracted from experimental and theoretical results previously derived, and these are summarized in Table 2 [20,22]. The transmission dynamics are governed by the model system detailed in S1 Appendix. The model illustration is presented in Fig 1, the model variables and their initial values [20] are given in Table 1. The definition and numerical values of the model parameters are given in Table 2.

We use data from the 2017/2018 YF outbreak in Brazil [27] to calibrate the model parameters, via the method of least squares. Note that for this outbreak, we perform the model fitting

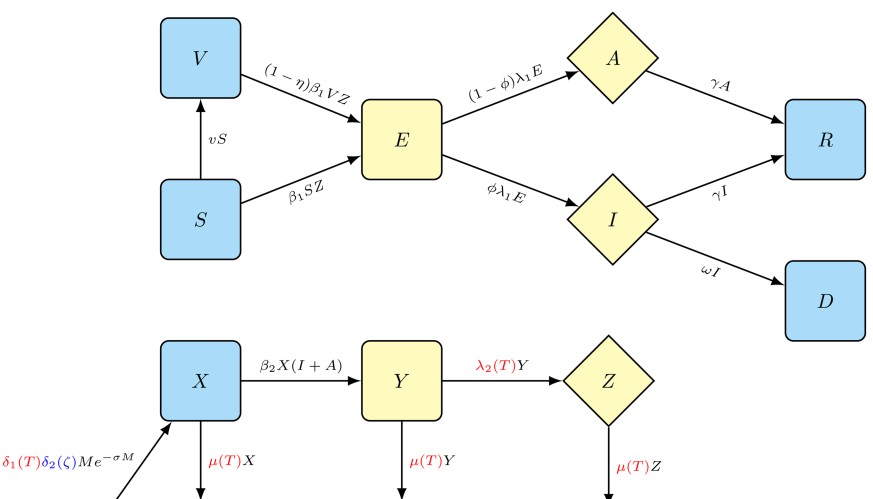

**Fig 1. An illustration of the compartmental model for the YF transmission dynamics: Yellow compartments represent the infected classes (including the exposed classes for human and mosquitoes), while the diamond-shaped compartments are for the infectious classes.** Red terms indicate temperature-dependent model parameters while blue terms indicate rainfall-dependent model parameters.

**Table 1. Model variable descriptions and initial conditions: Brazil 2017/2018 outbreak.**

| State Variable | Definition | Initial Value [20] |
|---|---|---|
| $S$ | Susceptible Humans | 41,893,754 |
| $V$ | Vaccinated Humans (Emergency Vaccination) | 0 |
| $E$ | Exposed Humans | 0 |
| $I$ | Infectious Symptomatic Humans | 21 |
| $A$ | Infectious Asymptomatic Humans | 25 |
| $R$ | Recovered Humans | 167,575,200 |
| $D$ | Disease-Induced Dead Humans | 0 |
| $X$ | Susceptible Mosquitoes | 418,937,761 |
| $Y$ | Exposed Mosquitoes | 150 |
| $Z$ | Infectious Mosquitoes | 995 |

**Table 2. Biological explanations and values of the model parameters.**

| Parameter | Definition | Value | Source |
|---|---|---|---|
| $\beta_1$ | Mosquito to human transmission rate | $4.789 \times 10^{-8}$ infections/week | estimated |
| $\beta_2$ | Human to mosquito transmission rate | $1.312 \times 10^{-10}$ infections/week | estimated |
| $\phi$ | Fraction of infected humans that develop symptoms | 0.45 | [10] |
| $\omega$ | Human death rate | 2.38 week$^{-1}$ | [23] |
| $\lambda_1^{-1}$ | Human intrinsic incubation period | 1.31 weeks | [24] |
| $\lambda_2(T)^{-1}$ | Extrinsic mosquito incubation period | see equation S1_(5) | [25] |
| $\gamma^{-1}$ | Duration of infectiousness | 0.82 weeks | [26] |
| $\mu(T)$ | Mosquito death rate | see equation S1_(4) | [22,25] |
| $\delta_1(T)\delta_2(\zeta)$ | Mosquito development rate | see equation S1_(3) | [20,22] |
| $\sigma$ | Density-dependency strength (mosquito reproduction) | $5.8 \times 10^{-10}$ | estimated |
| $\eta$ | Vaccine Efficacy | 0.95 | assumed |

accordingly to estimate the transmission parameters $\beta_1$ and $\beta_2$ under the average temperature ($\approx 24\,°C$) [20] and rainfall ($\approx 28$ mm weekly) [28] values during the 2017/2018 outbreak. The comparison of our model projection to the data is shown in Fig 2. We observe that the Root Mean Squared Error (RMSE) of our fit is 31.6 infections. This level of error reflects minor discrepancies around the outbreak peak but does not compromise the ability of the model to capture the overall shape, timing, and qualitative behavior of the epidemic curve.

We consider how a hypothetical outbreak would look under different climate scenarios: RCP 4.5 and RCP 8.5 in 2050 and analyse the impact of mosquito control and emergency vaccination rates could impact future outbreaks Different emergency vaccination rates are considered in response to the different RCP scenarios. We assess the impact of the different rates by simulating the projected cumulative cases and the basic human infection reproduction number under different vaccination regimes.  We simulate the impact of a hypothetical future outbreak under different temperature and rainfall values (which correspond to different RCP scenarios as described in Veiga, S. F et al.[28]) by using Python, primarily with the **numpy** and **scipy** packages. Model fitting was done by using the method **scipy.optimize** and simulations were run by using the methods **scipy.solve_ivp, scipy.odeint**.

The outbreak potential and intensity can be quantified by the basic human infection reproduction number, $\mathcal{R}_H$, which gives the average number of infected humans produced by the introduction of a single infected mosquito in the population [29]. There will be a disease outbreak if and only if $\mathcal{R}_H > 1$, so the goal of avoiding an outbreak is to implement the interventions that lead to $\mathcal{R}_H < 1$ [29]. The explicit formula we derived for $\mathcal{R}_H(T)$ is given below, from

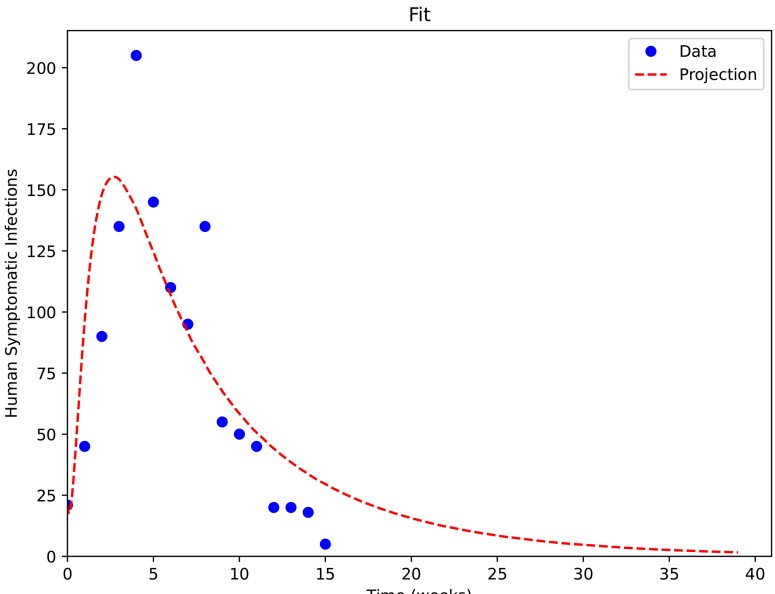

**Fig 2. Simulated curve of symptomatic infectious humans (red), from the fitted YF model, compared to the incidence data from the 2017/2018 outbreak (blue dots).**

which we calculated $\mathcal{R}_H$ = 5.36 for the 2017/2018 Brazil outbreak.

$$\mathcal{R}_H(T) = \frac{\beta_1 S_0 \lambda_2(T)}{\mu(T)(\mu(T) + \lambda_2(T))}. \tag{1}$$

A detailed explanation of each term in the expression for $\mathcal{R}_H$ is provided in S1 Appendix.

## Results

### Cumulative cases

In Figs 3 and 4, we plot the cumulative symptomatic infections under different temperature and rainfall scenarios. First, we set the rainfall value to a constant 28 mm per week and simulate the outbreak for different temperature values between 20.1 °C and 28.1 °C. These simulations results show that increasing average temperatures lead to an increase in the cumulative number of cases in our study region that is already an endemic zone. In particular, over the course of the first 15 weeks of the outbreak, approximately 4500 cumulative symptomatic infections are expected under the historically observed average temperature of 24.1 °C. Increasing the average temperature to 28.1 °C yields an increase in the total number of cumulative symptomatic infections to approximately 5000 over the course of the first 15 weeks. Over the course of the year (52 weeks), the simulations project an increase in cases of 23.6 % when the average temperature increases from 24.1 °C to 28.1 °C. We also note that under theoretically more extreme temperature values, i.e., greater than 35 °C (which is not projected for the region under consideration), the cumulative infections will decrease due to less favourable mosquito survival conditions.

Next, we simulate the outbreak under different weekly rainfall values, ranging from 24 mm to 32 mm, by holding temperature constant at a value of 24.1°C. Increasing rainfall values

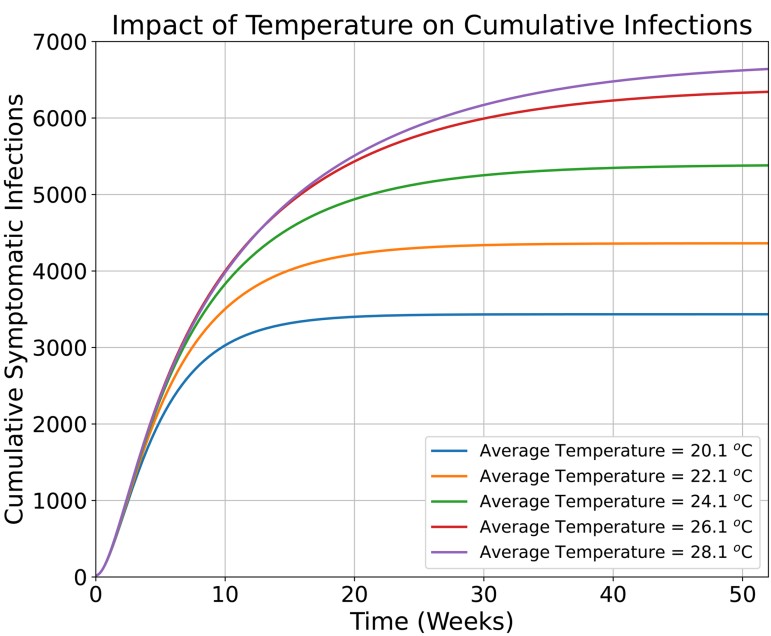

**Fig 3. Impact of temperature on the cumulative cases in the study region, where average rainfall is held constant at a value of approximately 28 mm.**

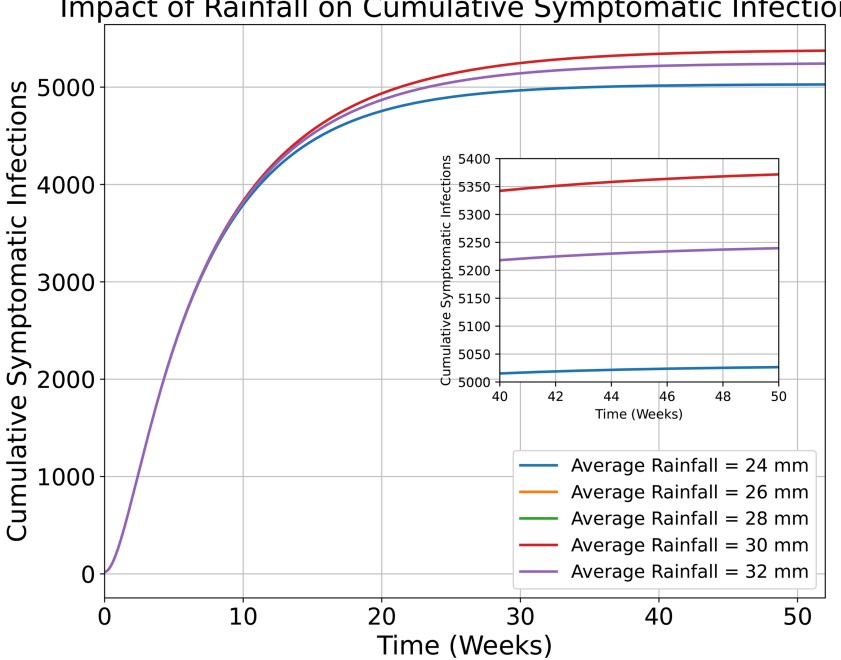

**Fig 4. Impact of rainfall on the cumulative cases in the study region, where average temperature is held constant at a value of approximately 24 ºC.**

leads to increases in the cumulative number of cases until reaching a critical value of approximately 29 mm. Then, if average weekly rainfall values increase further, we project a decrease

in the cumulative number of cases, due to less hospitable mosquito breeding conditions under this extreme rainfall condition. As can be seen from Fig 4, under the range of rainfall values between 24 mm to 32 mm, the cumulative symptomatic infections are expected to remain within the range of 5000 to 5400 over the course of one year. While the change in these values is not as extreme as the changes which result from increasing temperatures, they are still significant enough to warrant investigation. Furthermore, these values only reflect the impact of different average rainfall values under a fixed average temperature of approximately 24 °C. Harsher RCP scenarios typically result in an increase in both temperature and rainfall values in the region of study.

The joint impact of temperature and rainfall is shown in the heatmap of cumulative symptomatic infections (Fig 5), where the expected cumulative numbers of cases projected by the simulations under RCP 4.5 and RCP 8.5 scenarios are marked. For the study region, the cumulative cases will increase from the 2017/2018 level by 7.7 % and 8.1 % respectively under RCP 4.5 and RCP 8.5 scenarios. Finally, we also note that changes in temperature seem to be significantly stronger in affecting cumulative infections in comparison with the change of rainfall, in the ranges examined.

## Reproduction number and preventing an outbreak

Fig 6 plots the variation of the basic human infection reproduction number, $\mathcal{R}_H$, when temperature varies, with the help of the analytic formula (Eq 1). The plot shows that temperature plays a key role in the change of $\mathcal{R}_H$. Under the harsher RCP 8.5 scenario, the average number of human infections produced by an infected mosquito in the study region is expected to

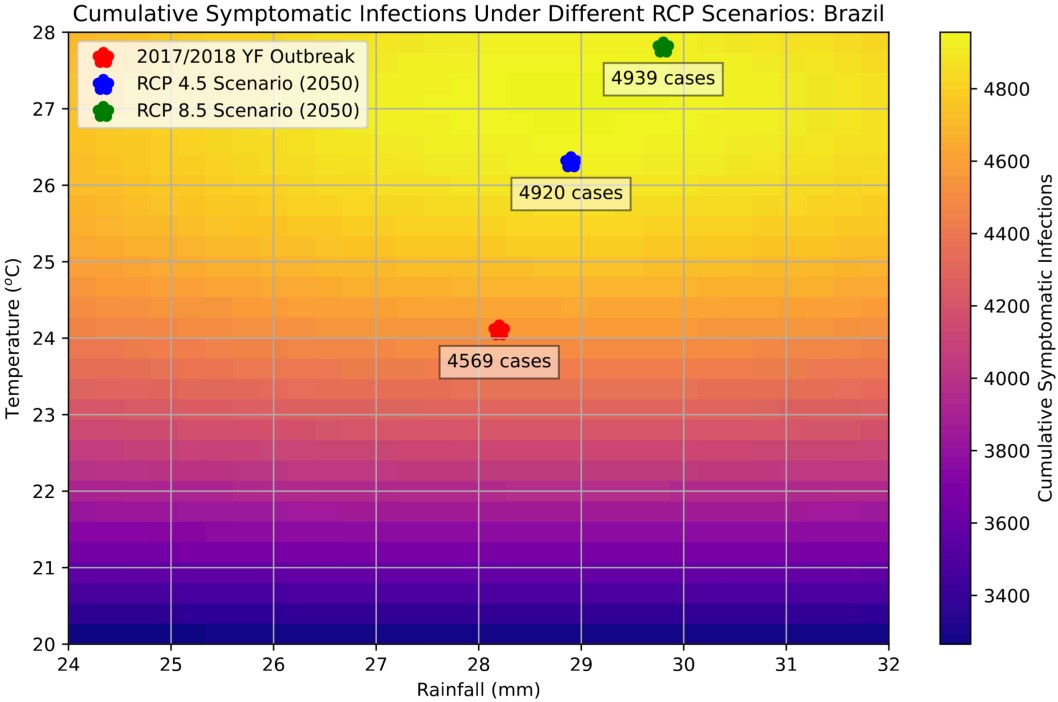

**Fig 5. Impact of temperature and rainfall on cumulative symptomatic infections in the study region under different RCP scenarios, over the course of the first 15 weeks of the outbreak.**

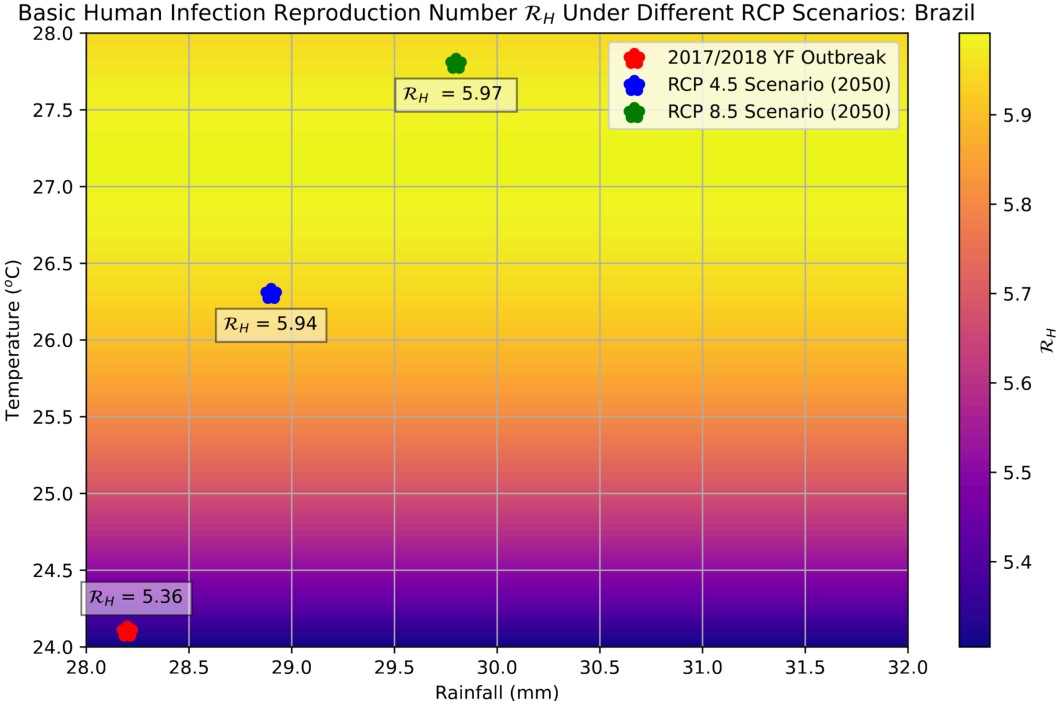

**Fig 6. The heatmap of the basic human infection reproduction number in the study region by varying the average temperature and average rainfall in the range under different climate projection scenarios.**

increase from 5.36 to 5.97—this increase highlights the public health need for taking actions to mitigate climate change.

In the absence of actions to alter the projected climate scenarios, significant public health efforts are required to prevent an outbreak. To quantify the scale of these efforts, we use the analytic formula Eq 1 to generate the threshold curve $\mathcal{R}_H = 1$ shown in Fig 7. Recall that $\beta_1$ represents the mosquito-to-human transmission rate. This parameter is directly influenced by public health interventions: reductions in $\beta_1$ can be achieved through vector control measures aimed at minimizing human-mosquito contact, as well as social interventions such as risk communication and the promotion of personal protection measures [30]. To prevent an outbreak, the value of $\beta_1$ must fall below the threshold curve in Fig 7. However, given our data-fitted estimate of the current transmission rate ($\beta_1 = 4.789 \times 10^{-8}$ infections/week), it is already very difficult, if not impossible, to avert the 2017/2018 outbreak under current climate conditions by relying solely on reducing human-mosquito contact.

### The need and effect of additional emergency vaccination

We have illustrated the importance of several key model parameters relevant to the temperature, rainfall and mosquito-to-human transmission rate on the cumulative infections and the basic human infection reproduction number in the study region. The simulations highlight the value of non-pharmaceutical interventions, alongside routine vaccination, in reducing transmission. However, under projected climate conditions, even stricter control measures may be necessary to prevent future outbreaks.

Under the assumption that an outbreak is unavoidable, another possible choice of public health intervention is the administration of an emergency vaccine. This motivates our next

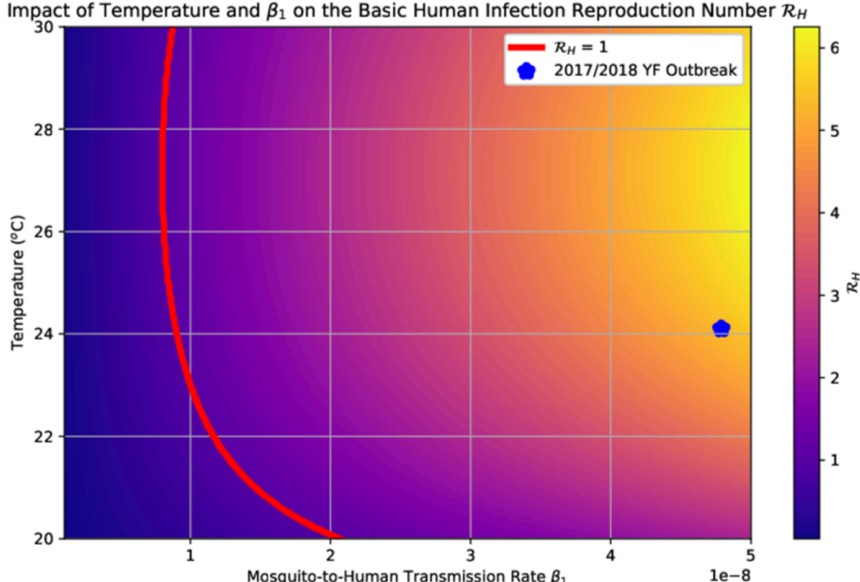

**Fig 7. The basic human infection reproduction number in the study region under different temperatures and mosquito-to-human transmission rates.** The red curve is where $\mathcal{R}_H = 1$, to the left of which an outbreak can be averted.

set of simulations and analyses, shown in Fig 8. In designing these simulations, we assume that the emergency vaccine is administered shortly after the beginning of the outbreak and becomes effective (with 95 % efficacy) after 10 days. Therefore, an individual enters in the $V$ compartment 10 days after vaccination and remains in the compartment hereafter.

In simulations summarized in Fig 8, we consider different vaccination rates, $v$, corresponding to the fraction of the susceptible population that becomes effectively vaccinated on a weekly basis. The top-left panel provides simulated projections for the number of infectious (both asymptomatic and symptomatic) humans. The top-right panel provides the number of both asymptomatic and symptomatic cumulative infections. The bottom-left panel gives the number of symptomatic human infections. The bottom-right panel provide the number of symptomatic cumulative infections. Our results showcase the utility of vaccination as a mitigating measure, yielding significant decrease in cumulative infections as vaccination rates are increased.

In S2 Appendix, we show the number of humans who have entered the emergency vaccinated compartment following administration of the emergency vaccination program. This gives the number of the emergency vaccine doses administered.

We then consider the impact of emergency vaccination under the harsh RCP 8.5 climate scenario projection in Fig 9. Without the administration of additional emergency vaccines, the cumulative infections are projected to increase by approximately 8 % under RCP 8.5 when compared to the 2017/2018 outbreak. If an emergency vaccination rate of $v = 0.08$ is used as a mitigating measure, the cumulative cases will only be increased by 4.9 % under the RCP 8.5 scenario.

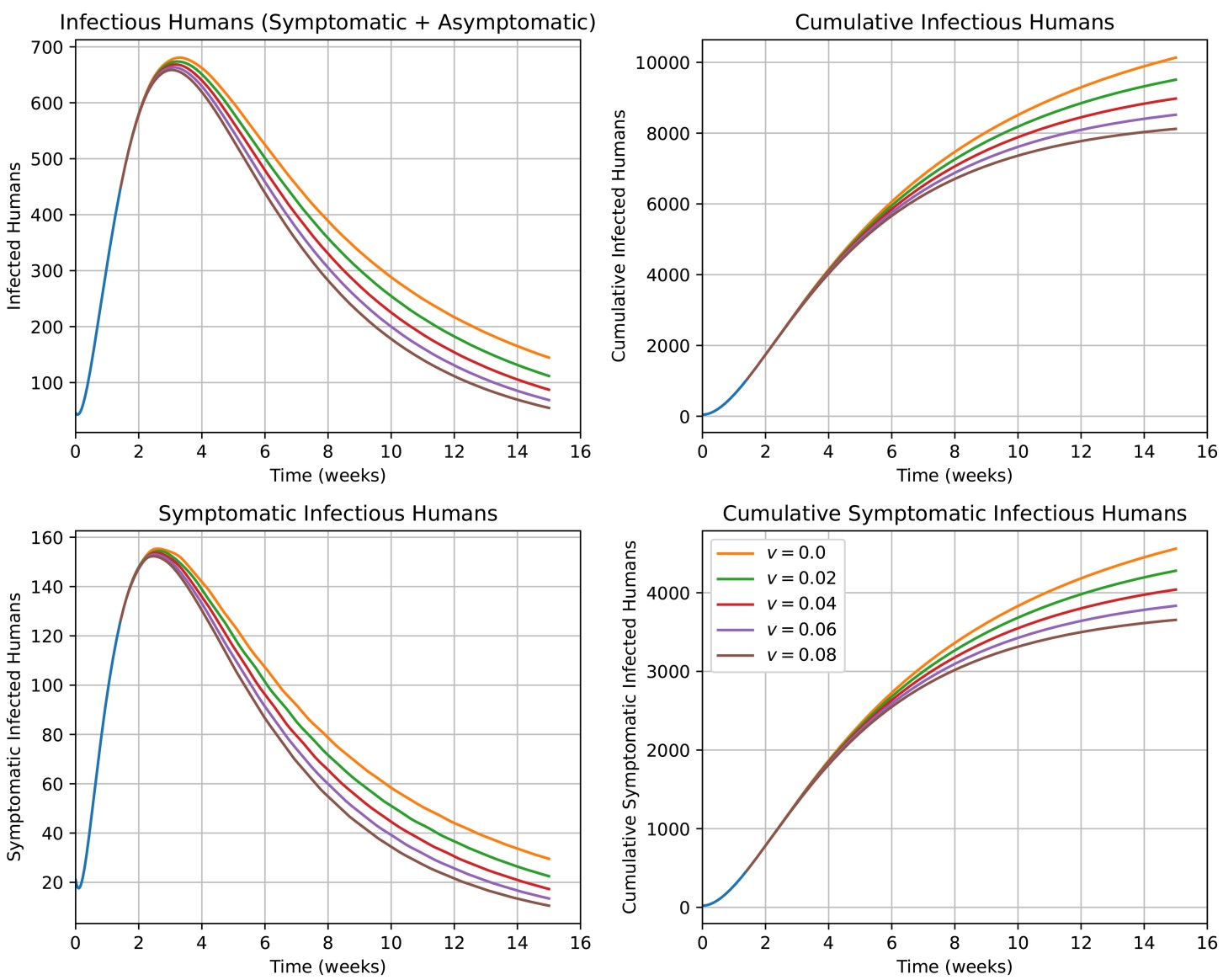

**Fig 8. The impact of a (hypothetical) administration of emergency vaccine with different vaccination rates on the incidence and cumulative cases during the 2017/2018 outbreak.** Different $v$ values correspond to different vaccination rates.

### The public health impact of emergency vaccination

The public health impact of emergency vaccination can be inferred by comparing it to the level of reduction of the mosquito-to-human transmission rate, $\beta_1$, that is equivalent to a given vaccination rate. This is illustrated in Fig 10 for the projection under the RCP 8.5 scenario, and in Fig B in S2 Appendix, for the 2017/2018 outbreak. The blue curve describes the cumulative infections while varying the transmission rate $\beta_1$ in the absence of an administration of emergency vaccine ($v = 0$); while the horizontal lines show the cumulative infections corresponding to different emergency vaccination rates, under the estimated transmission rates. From Fig 10, we note that the cumulative human infections can be reduced to 8519 by the administration of emergency vaccination at the rate of $v = 0.08$ while maintaining the

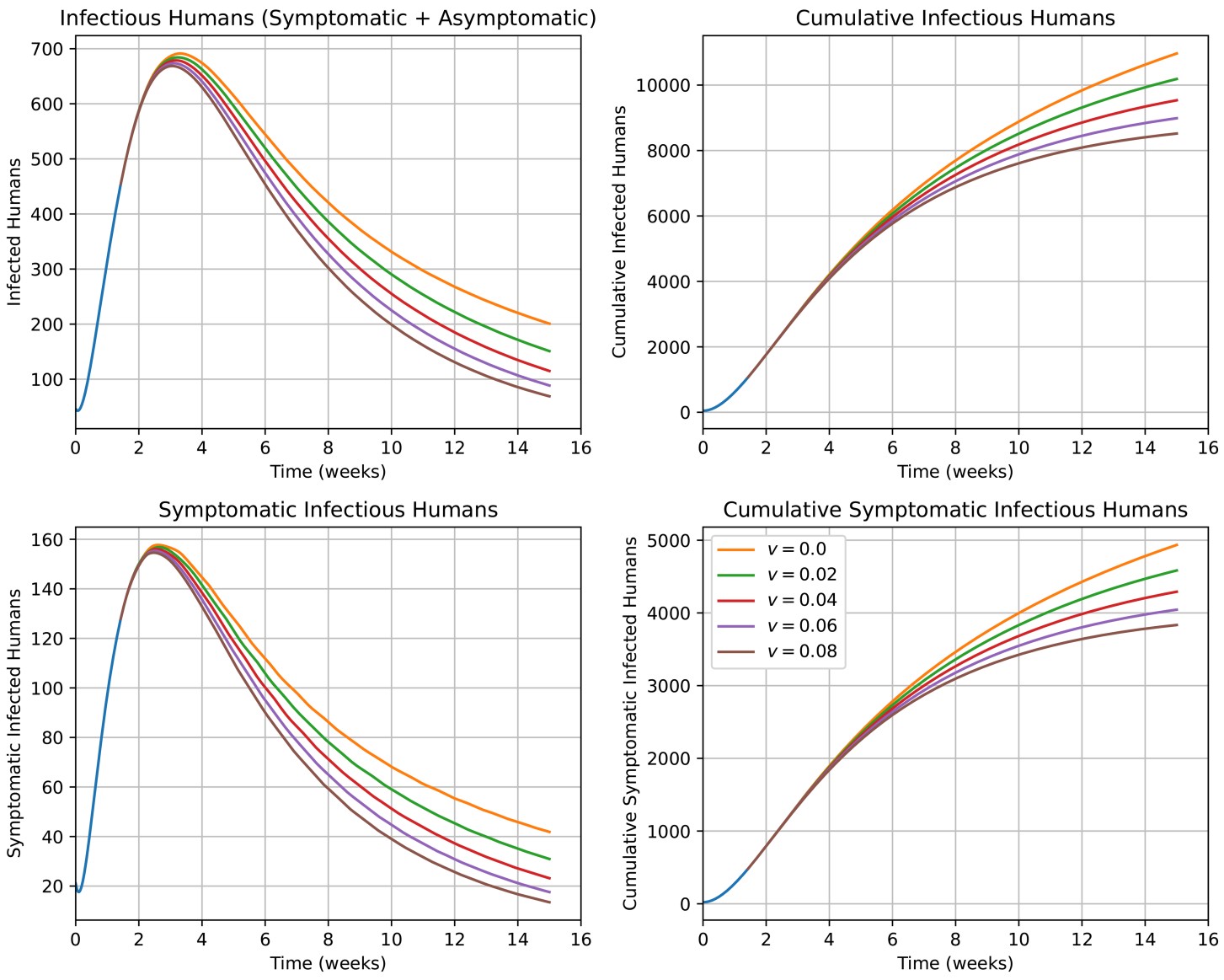

**Fig 9. The impact of administration of emergency vaccine with different vaccination rates on the incidence and cumulative cases under RCP 8.5 scenario.**

mosquito-to-human contact $\beta_1 = 4.789 \times 10^{-8}$ infections/week. If we want to achieve this level of reduction of cumulative infections solely by reducing the mosquito-to-human contacts (without using emergency vaccine), we must reduce these contacts from the 2017/2018 level by 23.0 %.

The heatmap in Fig 11 demonstrates the role of emergency vaccination in an integrative approach towards mitigating future YF outbreaks exacerbated by the RCP 8.5 projected climate conditions. The purple curve gives the condition for the mosquito-to-human transmission rate $\beta_1$ and the emergency vaccination rate $v$ to ensure the cumulative infections will not exceed the 2017/2018 outbreak level. The orange curve shows values of the emergency vaccination rate and the mosquito-to-human transmission rate which will yield a 25 % reduction in cumulative infections compared to the 2017/2018 outbreak. For example, a combination of

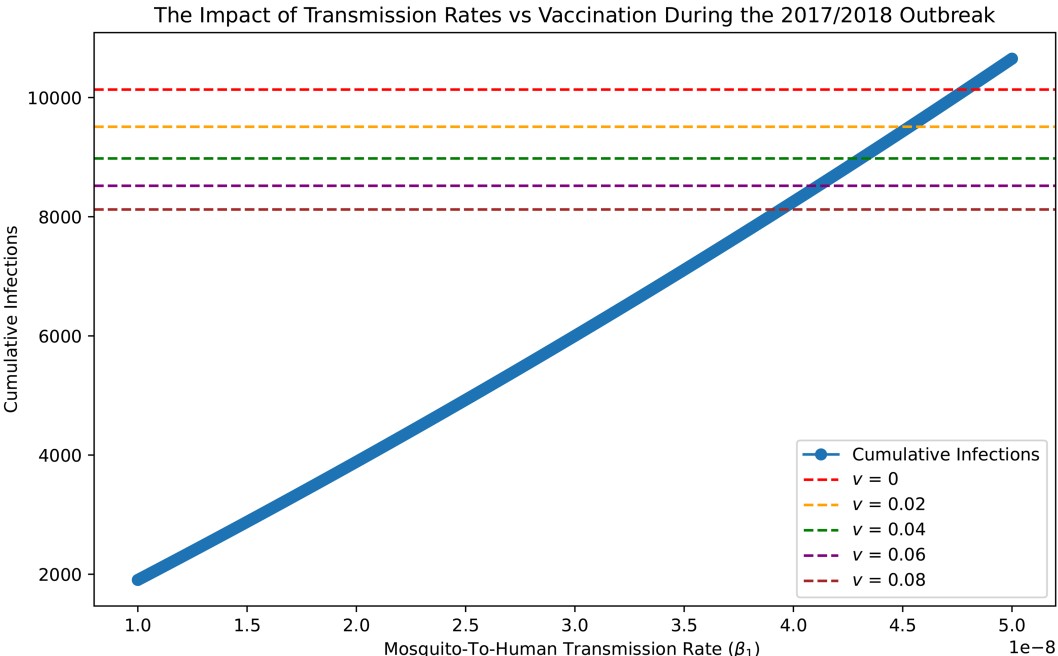

**Fig 10. The equivalence between reducing the mosquito-to-human transmission rate and administrating emergency vaccine in terms of reducing the cumulative (both symptomatic and asymptomatic) infections of the first 15 weeks from epidemic outbreak in the study region under the RCP 8.5 scenario.**

$v = 0.03$ and $\beta_1 = 4 \times 10^{-8}$ will ensure the cumulative cases under RCP 8.5 scenario be 75% of the cumulative infections in the 2017/2018 outbreak. Similarly, the red curve shows the values of these parameters which will yield a 50 % reduction compared to the 2017/2018 outbreak.

## Discussion

We made projections regarding the impact of future climate on YF infections, using a case study for a region with reported YF outbreaks. The impact was assessed with the help of established relationship between key model parameters and temperature and rainfall. The calibrated model was used to examine the need and effect of plausible mitigation measures including the administration of emergency vaccination to mitigate future YF outbreaks exacerbated by the projected climate conditions.

Our results suggest that in some regions where YF is endemic, harsher climate conditions such as those projected under RCP 4.5 and RCP 8.5 scenarios will result in increases in the basic human infection reproduction number and the cumulative infections by 2050. We used a model framework similar to that developed in Sadeghieh et al., but we found that harsher RCP scenarios will lead to increasing cumulative cases in the study region, as our analyses showed that the projected harsher climate conditions (i.e., RCP4.5 and RCP8.5) remain favorable for the overall mosquito production, development and survival [20]. Another important question is whether climate change will lead to increases in the cumulative cases and basic reproduction number in more temperate regions, but more detailed quantitative analyses will have to be left for future investigations incorporating the demographic, epidemiological, and environmental details in the regions under consideration.

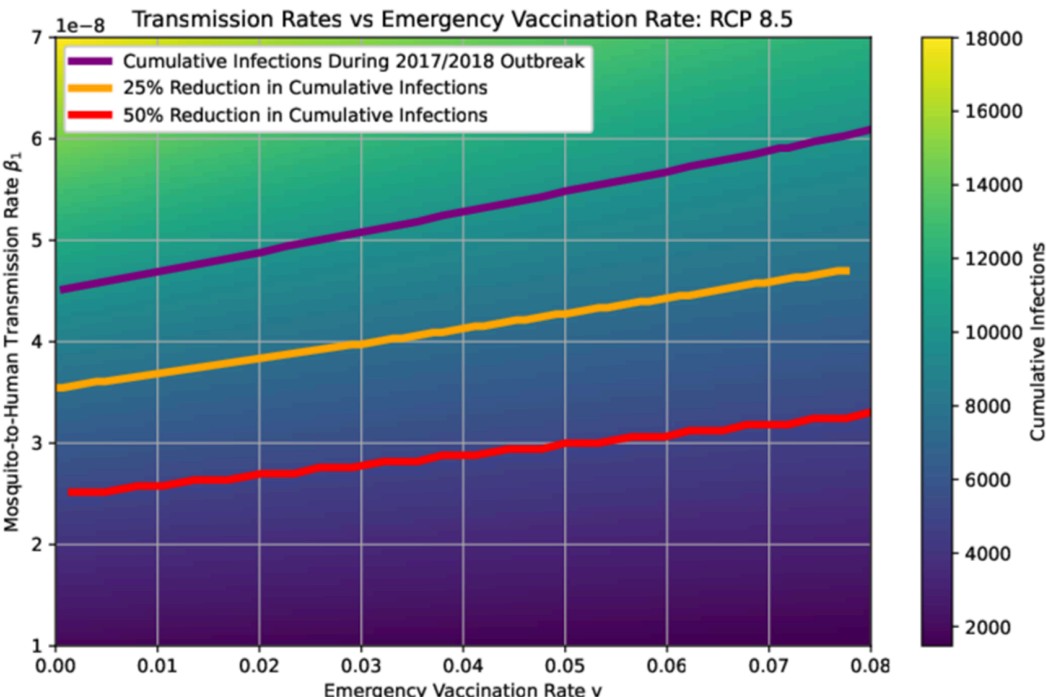

**Fig 11. The impact of the mosquito-to-human transmission rate and the emergency vaccination rate on the reduction of cumulative infections (both symptomatic and asymptomatic) during the first 15 weeks of the epidemic in the study region, under the RCP 8.5 scenario, assuming a constant emergency vaccination rate throughout the outbreak.**

Prior to the outbreak, the states of Rio de Janeiro and São Paulo had already vaccinated about 13.2 million people. During the campaign, an additional 13.3 million people were vaccinated in São Paulo, 6.5 million in Rio de Janeiro and 1.85 million in Bahia. This resulted in a vaccination coverage of 53.6 %, 55.6 %, and 55.0 % respectively and across all 77 municipalities with the greatest risk of yellow fever [31]. It must be noted that in the state of São Paulo alone, the risk areas for YF expanded within a very short period of time, which tripled the target population for vaccination from 10 to 30 million. The sudden nature of this expansion, mostly affecting municipalities with low vaccination coverage rate or without any previous recommendation for routine vaccination, created significant challenges to start emergency vaccination, as there was neither enough time nor vaccine doses to cover the entire population [32]. The impact of vaccination cannot be understated and was one of the key considerations in this work. In particular, we investigated the impact of additional emergency vaccination, with results to demonstrate the public health value of the emergency vaccines. We suggest that these results—in addition to routine YF vaccination—should be considered by policy decision makers in their strategic planning to mitigate the worsening YF transmission situation given the projected increase of both temperature and rainfall. The model analysis on the necessary additional emergency vaccination rate in any integrative prevention and control plan provides information to guide logistic implementation.

The importance of meteorological factors in vector-borne disease outbreaks and the impact of climate change on the range expansion of vector-borne diseases have been considered on a global scale. Gaythorpe et al. estimated 25 % more YF deaths by 2050 under an RCP 8.5 scenario [25]. In our work, we estimate an increase of 8.1 %, during the first 15 weeks, in the

total number of symptomatic cases during 2050 under an RCP 8.5 scenario, compared to the baseline (2017/2018 outbreak). Differences could be explained by the setting (Africa versus Latin America). However, our prediction is also sensitive to the duration of epidemics. When the impact is evaluated over a year, a temperature increase of 4 degrees leads to a projected 23.6 % increase in the number of infections, which aligns more closely with the results of Gaythorpe et al.

The impact of climate factors such as temperature or rainfall on other mosquito-borne diseases has also been well studied. For example, Wu et al. concluded that temperature and rainfall are amongst the most crucial parameters to be considered in modelling the dynamics of vector-borne diseases in China [33]. In particular, this study projected that malaria incidence in Northern China will increase from 69% to 182% by 2050 [34]. These trends are also reflected in our projections of YF cases in Brazil by 2050, under harsher temperature and rainfall conditions. Iwamura et al. found consistent results for invasive mosquitoes with the potential to carry arboviruses: their modelling-based analyses suggest that extended periods of mosquito development, due to more suitable climate conditions, are expected to accelerate the global potential for mosquito invasion [35].

One key limitation of this modelling is the method with which additional emergency vaccination is implemented. In particular, this represents a hypothetical scenario in which additional emergency vaccines are distributed in addition to the natural immunity and vaccination coverage. As in Sadeghieh et al., the initial 80% immune fraction takes into account the historical response to the 2017/2018 outbreak, and represents a strong response by health authorities [20]. In our modelling, another limitation of considering further/additional vaccination is that the rate of vaccination is assumed constant. While this can give a good estimate regarding volume of additional vaccines required, some realism is lost as vaccines are typically not distributed at a constant rate in response to an outbreak. Moreover, the model does not directly model the historical vaccination campaign. While this provides us a convenient and useful way to model the impact of different vaccination rates, it should still be noted as a limitation. Nevertheless, such modelling has previously appeared in the existing literature and can provide important insights regarding vaccine distribution [36]. In particular, our model considered this hypothetical vaccination under different RCP scenarios and provided an estimate on how varying climate patterns can be expected to affect the need for additional vaccines in the future. It should also be noted that the use of (constant) average temperature and rainfall values in the region rather than seasonally changing values is a limitation. The work presented in this manuscript can be built upon by considering temperature and rainfall functions that vary with time. All of these limitations should be addressed in future work.

As noted in Kraemer et al., and supported by our results, it is also critical to explore additional avenues for disease mitigation in response to rising global temperature and precipitation changes [37]. The trends of global urbanization, leading to increased mosquito proliferation, should also be considered under different RCP scenarios. Kraemer et al. considered this impact via probabilistic machine learning modelling, providing further mechanistic evidence about the impact of human spread [37]. This human mobility is implicitly incorporated in our modelling study in a given region by adjusting the initial value of human population, for example, by decreasing the initial immunization coverage with increased importation of susceptible individuals, and increasing the initial values of infected population with importation of infections. In future work, global connectedness and urbanization should be further incorporated in an appropriately modified version of our model (meta-population model) to deal with YF spread on an international scale. Moreover, to better capture the spread of YF, it is important to also consider vertical transmission of YF from infected mosquitoes to their offspring [38]. Other factors that warrant consideration include the evolution of mosquitoes'

pesticide resistance and increasing human birth rates. Furthermore, migration rates of individuals into countries in which YF is not endemic should be considered. Such countries typically have highly susceptible populations (as no routine YF vaccination is recommended in non-endemic countries). An example of one such case is when travelers returned to China from Angola, leading to the first reported case of YF in China in 2016 [39]. These considerations are left for future modelling investigations.

## Supporting information

**S1 Appendix. The mathematical model and basic reproduction number derivation.**
(PDF)

**S2 Appendix. Supplementary figures.**
(PDF)

**S3 Appendix. Mathematical proof of well-posedness of the differential equation model.**
(PDF)

**S1 Code. S2 Fig A**: The impact of different vaccination rates ($v$) on the cumulative emergency vaccination doses require over the course of 15 weeks. **S2 Fig B**: The equivalence between reducing the mosquito-to-human transmission rate and administering additional emergency vaccines under the conditions of the 2017/2018 outbreak.
(IPYNB)

## Author contributions

**Conceptualization:** Tedi Ramaj, Xiaotian Wu, Marco Tosato, Fernando Morelli, Yael Thollot, Edith Langevin, Edward Thommes, Woldegebriel Assefa Woldegerima, Jianhong Wu.

**Data curation:** Tedi Ramaj, Fernando Morelli, Yael Thollot, Edith Langevin, Edward Thommes, Jianhong Wu.

**Formal analysis:** Tedi Ramaj, Xiaotian Wu, Marco Tosato, Fernando Morelli, Yael Thollot, Edith Langevin, Edward Thommes, Woldegebriel Assefa Woldegerima, Jianhong Wu.

**Funding acquisition:** Fernando Morelli, Yael Thollot, Edith Langevin, Edward Thommes, Jianhong Wu.

**Methodology:** Tedi Ramaj, Xiaotian Wu, Marco Tosato, Fernando Morelli, Yael Thollot, Edith Langevin, Woldegebriel Assefa Woldegerima, Jianhong Wu.

**Project administration:** Jianhong Wu.

**Software:** Tedi Ramaj, Marco Tosato.

**Supervision:** Xiaotian Wu, Fernando Morelli, Yael Thollot, Edith Langevin, Edward Thommes, Woldegebriel Assefa Woldegerima, Jianhong Wu.

**Validation:** Tedi Ramaj, Xiaotian Wu, Marco Tosato, Jianhong Wu.

**Visualization:** Tedi Ramaj, Marco Tosato, Jianhong Wu.

**Writing – original draft:** Tedi Ramaj, Xiaotian Wu, Marco Tosato, Jianhong Wu.

**Writing – review & editing:** Tedi Ramaj, Xiaotian Wu, Marco Tosato, Fernando Morelli, Yael Thollot, Edith Langevin, Edward Thommes, Woldegebriel Assefa Woldegerima, Jianhong Wu.

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
