## [Decision Letter · Decision Letter 0]

2 Apr 2025

PNTD-D-25-00305Risk Assessment and Mitigation Evaluation of Future Yellow Fever Outbreaks Under Different Climate Scenarios: Insight from a Case Study of BrazilPLOS Neglected Tropical Diseases Dear Dr. Ramaj, Thank you for submitting your manuscript to PLOS Neglected Tropical Diseases. After careful consideration, we feel that it has merit but does not fully meet PLOS Neglected Tropical Diseases's publication criteria as it currently stands. Therefore, we invite you to submit a revised version of the manuscript that addresses the points raised during the review process. Please submit your revised manuscript within 30 days Jun 01 2025 11:59PM. If you will need more time than this to complete your revisions, please reply to this message or contact the journal office at plosntds@plos.org. Please include the following items when submitting your revised manuscript: * A rebuttal letter that responds to each point raised by the editor and reviewer(s). You should upload this letter as a separate file labeled 'Response to Reviewers'. This file does not need to include responses to any formatting updates and technical items listed in the 'Journal Requirements' section below. * A marked-up copy of your manuscript that highlights changes made to the original version. You should upload this as a separate file labeled 'Revised Manuscript with Track Changes'. * An unmarked version of your revised paper without tracked changes. You should upload this as a separate file labeled 'Manuscript'. If you would like to make changes to your financial disclosure, competing interests statement, or data availability statement, please make these updates within the submission form at the time of resubmission. Guidelines for resubmitting your figure files are available below the reviewer comments at the end of this letter. We look forward to receiving your revised manuscript. Kind regards, Helton C. Santiago, M.D., Ph.DAcademic EditorPLOS Neglected Tropical Diseases Audrey LenhartSection EditorPLOS Neglected Tropical Diseases

Shaden Kamhawi

co-Editor-in-Chief

Paul Brindley

co-Editor-in-Chief

**Additional Editor Comments:** The manuscript was evaluated by three specialists who point some interesting minor points to be fixed. We are returning the manuscript to the authors so they may evaluate the suggestions and amend the paper in the points they consider relevant. **Journal Requirements:**

At this stage, the following Authors/Authors require contributions: Tedi Ramaj, Xiaotian Wu, Marco Tosato, Fernando Morelli, Yael Thollot, Edith Langevin, Edward Thommes, Woldegebriel Assefa Woldegerima, and Jianhong Wu. Please ensure that the full contributions of each author are acknowledged in the "Add/Edit/Remove Authors" section of our submission form.

4) We notice that your supplementary Figures, and Tables are included in the manuscript file. Please remove them and upload them with the file type 'Supporting Information'. Please ensure that each Supporting Information file has a legend listed in the manuscript after the references list.

5) We note that your Data Availability Statement is currently as follows: "All relevant data has been cited within the paper and may be found in the corresponding references.". Please confirm at this time whether or not your submission contains all raw data required to replicate the results of your study. Authors must share the “minimal data set” for their submission. PLOS defines the minimal data set to consist of the data required to replicate all study findings reported in the article, as well as related metadata and methods (https://journals.plos.org/plosone/s/data-availability#loc-minimal-data-set-definition).

- The points extracted from images for analysis..

6) Please ensure that the funders and grant numbers match between the Financial Disclosure field and the Funding Information tab in your submission form. Note that the funders must be provided in the same order in both places as well.

**Reviewers' comments:** Reviewer's Responses to Questions

**Key Review Criteria Required for Acceptance?**

**Methods:**

-Are the objectives of the study clearly articulated with a clear testable hypothesis stated?

-Is the study design appropriate to address the stated objectives?

-Is the population clearly described and appropriate for the hypothesis being tested?

-Is the sample size sufficient to ensure adequate power to address the hypothesis being tested?

-Were correct statistical analysis used to support conclusions?

-Are there concerns about ethical or regulatory requirements being met?

Reviewer #1: (No Response)

Reviewer #2: Introduction:

-Great job introducing the problem and identifying how YF can be impacted by climate change. You cite a lot of relevant literature for YF modeling and projections in Europe, Australia, and Canada. One point I would consider is if there are any examples of non-modeled increases in YF (or any other mosquito born pathogen) prevalence or impact due to climate change (i.e., actual data from South America) you can reference? For example, have case loads increased across the last 20 years or change their distributions to more temperate areas? Added in the references could help really nail down why this study is so important.

-I would also consider taking some space to explain what type or epidemiological model you are using and why it is appropriate in your introduction

-You could also use this space to highlight your incorporation of vaccination coverage in your model, and why that is both novel and important. Specifically, some points that you make in your discussion about vaccination coverage in this region prior to the outbreak could be moved here to give the reader a bit more context on this specific outbreak.

-Page 2 line 16- Define WHO and EYE acronyms

-Lines 28-29: Sentence “Certain regions currently endemic to YF….” Needs a citation.

Methods/Results:

-Some more detail is needed in your methods with regard to your model parameters. More accurately, the details are there, but need to be moved around. For example, you state that parameters v and n are either assumed or derived but you do not prompt the reader to go to Table 2 until the end of this section (i.e., lines 65-86). It is then unclear where the parameters were assumed or calculated from if you are just looking at the table. I would suggest clarifying early on in this section that all parameters were either derived from the case study outbreak (and cite where you got that data from), or were based on previously published literature on this disease system. Again, the details are there, they were just a little difficult to find at times.

-In that same vein, move the sentence from lines 131-134 (source of temp and rainfall projections) up to your methods

-Consider moving Fig 2 down to results and also reporting the performance of the fitted model. The figure is informative, but putting some numbers with it would send home the message that this model is well calibrated to the data.

-Inversely, sentences from lines 106-107 and 118-119 about how you simulated the outbreak under different temp and rainfall values may belong in your methods section.

Reviewer #3: The study aims to formulate a mathematical model of Yellow Fever transmission with a case study of the

region in southeastern Brazil. The authors validated the model using historical data, then ran simulations to generate projections of future outbreaks under different climate scenarios and evaluate the outcomes of different mitigation measures. The model parameters are appropriate and grounded in the literature. Data from Brazil's large 2017/2018 YF outbreak was used to calibrate the model parameters.

The authors use the framework of a standard vector-borne disease transmission dynamics model involving humans as hosts and mosquitoes as vectors.

**Results:**

-Does the analysis presented match the analysis plan?

-Are the results clearly and completely presented?

-Are the figures (Tables, Images) of sufficient quality for clarity?

Reviewer #1: (No Response)

Reviewer #2: Methods/Results:

-One questions that I have regarding your projections: Are the average temperatures and rainfall values that you are using averaged across the year or are they the average values for a specific season? You report increases in cases based by week of the outbreak, but if I am not mistaken, the temp and rainfall numbers for each scenario are static and used to calculate the mosquito metrics. Do I have that correct, or am I misreading? This was unclear to a certain extent. As a follow up to that, does YF have seasonal outbreak patterns in this region? How might climate change impact the seasonality of temp and rainfall in this region, and might that impact YF outbreaks? Assessing seasonality might be beyond the scope of this study, but it may be worth noting in your discussion.

-Lines 165-170 may belong in your methods

-Figure 9: What is the slope of the line for cumulative infections?

-Was there a specific software that you used to run your model? If so, specify it and any packages you used in your methods section.

Reviewer #3: The results are presented clearly. Graphs are appropriately used to show different projections of cases and Ro with varying climate parameters and mitigation strategies.

**Conclusions:**

-Are the conclusions supported by the data presented?

-Are the limitations of analysis clearly described?

-Do the authors discuss how these data can be helpful to advance our understanding of the topic under study?

-Is public health relevance addressed?

Reviewer #1: (No Response)

Reviewer #2: Discussion:

-Lines 226-231: This is a great detail that may be worth moving up to the introduction and/or being expanded in the discussion section. What are some challenges to emergency vaccination in this region? This could help tie in how your results are useful for public health officials.

-Lines 283-284: Sentence, “Such countries typically have highly susceptible populations” needs a citation.

Reviewer #3: Limitations of the model are appropriately acknowledged.

The formulated model might help understand the dynamics of the YF Epidemic in different climate conditions. It also helps public health officers estimate emergency vaccination's impact during an epidemic.

**Editorial and Data Presentation Modifications?**

Reviewer #1: (No Response)

Reviewer #2: General Comments:

-Is there a particular reason why you chose to use RCP’s over more contemporary SSPs in CMIP 6 for your future climate projections? If so, introduce this in your methods or even introduction.

- Make sure all of your methods are in the methods section and that all your results are in the results section. I found myself having questions about how you did some assessments (particularly the vaccine assessment) after reading your methods section, only to have those questions answered later on in the results. This just helps me as the reader have the full story before I move on into your results and discussion.

Reviewer #3: accept

**Summary and General Comments:**

Reviewer #1: Thank you for the opportunity to review this manuscript which is overall very well written, the results are clearly presented and of sound methodology. One omission from the manuscript as it stands is a hypothesis for why the increase in YF is found here, and does it change the shape of the epidemic curve? E.g., are increasing temperatures increasing vector development rates or is rainfall more related to vector breeding or higher humidity? Perhaps you cant answer this with your methodology, but it would be interesting to incorporate into the discussion.

Please find some specific minor feedback related to the text.

Line 4: perhaps change the word “species” to “genus”, as those you mention are not species.

Line 5: instead of “and jaundice when the liver is attacked.”, you could consider changing to “with jaundice when there is hepatic involvement.”

Line 6: “However, climate change in general and temperature increase in particular” could change to “However, elevated temperatures related to natural and anthropogenic climate change may create more ideal….” The term “climate change in general” is open to interpretation.

Line 15-18: Not a complete sentence: “Many international collaborative projects such as the WHO’s project EYE have been launched due to the global public health interest, and interdisciplinary and integrative approach has been called for due to the inherent complexity of the prevention and control of YF.” Consider: “Due to the inherent complexity of the prevention and control of YF and the global public health interest, interdisciplinary and integrative approach has been called for, including the WHO’s project EYE”. Is EYE an acronym, if so, spell out on first use. Additionally, this is the first time you mention the World Health Organization, so you also need to spell it out fully.

Line 32: Either remove the article or make it singular: “the YF public health burdens”

Line 36-38: Conditions are already favorable for Ae. agypti in Europe and the mosquito has been present in Europe before (and is present currently in Cyprus and Madeira), what is limited its spread in Europe is not the climate, it's the introduction of the mosquito at significant quantities that it can create established populations.

Line 55: Remove article: “by the climate change.” to “by climate change”

Line 84: “We use data from the 2017/2018 YF outbreak in Brazil” - it would be good to know what this data looked like in time and space?

Line 216 and 2020: “harsher conditions” I would be clearer about what you mean by that. Do you mean harsher climate conditions, or harsher CO2 concentrations, by harsher I am assuming you mean higher?

Line 251-252: do you mean “invasive mosquitoes with the potential to carry arboviruses” ?

Line 263: “does not take into account the historical vaccination campaign” surely this is a significant issue here as the vaccine provides immunity for a lifetime.

Line 270: Consider changing to “and precipitation changes”, precipitation will not increase in all areas and our uncertainty around future rainfall patterns and intensity are higher than temperature.

Reviewer #2: This study utilizes modified SEAIR models to assess the possible impacts of future climate change, via changes to average temperature and precipitation, on yellow fever cases using a case study from Brazil as the basis for their modeling framework. The manuscript is well written and presents the basis of its assessment and results well. The findings are of importance both for public health officials, as well as the broader disease modeling community. Their incorporation of emergency vaccine efficacy in different scenarios in suppressing future possible outbreaks is novel, and well presented. Nevertheless, there are some places where more detail is needed and there are a few things that I think could be moved around. Overall, the manuscript is in need of minor edits, but on the whole represents a good body of work.

Reviewer #3: The model was formulated using an appropriate framework based on valid parameters. The results help understand the dynamics of YF epidemics and the impact of mitigation strategies.

PLOS authors have the option to publish the peer review history of their article (what does this mean?). If published, this will include your full peer review and any attached files.

Reviewer #1: **Yes: **Gina E C Charnley

Reviewer #2: No

Reviewer #3: **Yes: **Alexandre Sampaio Moura

---

## [Decision Letter · Decision Letter 1]

7 Aug 2025

Dear Dr Ramaj,

We are pleased to inform you that your manuscript 'Risk Assessment and Mitigation Evaluation of Future Yellow Fever Outbreaks Under Different Climate Scenarios: Insight from a Case Study of Brazil' has been provisionally accepted for publication in PLOS Neglected Tropical Diseases.

Best regards,

Helton C. Santiago, M.D., Ph.D

Academic Editor

Audrey Lenhart

Section Editor

Shaden Kamhawi

co-Editor-in-Chief

Paul Brindley

co-Editor-in-Chief

Reviewer's Responses to Questions

**Key Review Criteria Required for Acceptance?**

**Methods**

-Are the objectives of the study clearly articulated with a clear testable hypothesis stated?

-Is the study design appropriate to address the stated objectives?

-Is the population clearly described and appropriate for the hypothesis being tested?

-Is the sample size sufficient to ensure adequate power to address the hypothesis being tested?

-Were correct statistical analysis used to support conclusions?

-Are there concerns about ethical or regulatory requirements being met?

Reviewer #2: (No Response)

**Results**

-Does the analysis presented match the analysis plan?

-Are the results clearly and completely presented?

-Are the figures (Tables, Images) of sufficient quality for clarity?

Reviewer #2: (No Response)

**Conclusions**

-Are the conclusions supported by the data presented?

-Are the limitations of analysis clearly described?

-Do the authors discuss how these data can be helpful to advance our understanding of the topic under study?

-Is public health relevance addressed?

Reviewer #2: (No Response)

**Editorial and Data Presentation Modifications?**

Reviewer #2: (No Response)

**Summary and General Comments**

Reviewer #2: I'd like to thank the authors for their thoughtful responses to my comments. I believe the manuscript is improved and could be accepted as is. There are a few minor suggestions below that I would recommend looking at, but they are all minor things. Excellent work, and thank you for the opportunity to review your work.

Lines 24-28: I would recommend integrating some citations in the section to support these statements. I realize that these statements are generally accepted as fact, but a few supporting references would add credence here.

Line 107: You're missing a period after "future outbreaks"

Lines 153-154: I would suggest reworking this sentence as it is a little confusing. I would suggest, "Finally, we noted that changes in temperature had a larger impact on cumulative infections than changes in rainfall." The phrase "significantly stronger" would necessitate a statistical value in most cases.

Lines 198-199: Would it be possible for your to put a monetary or population value on the amount of vaccination required to mitigate the increase in cases here in the absence of climate change mitigation measure? This might be a good supporting metric for your discussion section. Fore example, "vaccination rate of 0.4 costs x amount, and increasing vaccination to the threshold necessary to mitigate the climate change caused increase (0.8) would cost x."

Lines 236-238: Integrate supporting citations here for these numbers.

PLOS authors have the option to publish the peer review history of their article (what does this mean?). If published, this will include your full peer review and any attached files.

Reviewer #2: No

---

## [Editor Report · Acceptance letter]

Dear Dr Ramaj,

We are delighted to inform you that your manuscript, " 

Risk Assessment and Mitigation Evaluation of Future Yellow Fever Outbreaks Under Different Climate Scenarios: Insight from a Case Study of Brazil," has been formally accepted for publication in PLOS Neglected Tropical Diseases.

Best regards,

Shaden Kamhawi

co-Editor-in-Chief

Paul Brindley

co-Editor-in-Chief
